# Machine Learning Methods to Predict Social Media Disaster Rumor Refuters

**DOI:** 10.3390/ijerph16081452

**Published:** 2019-04-24

**Authors:** Shihang Wang, Zongmin Li, Yuhong Wang, Qi Zhang

**Affiliations:** 1Business School, Sichuan University, Chengdu 610064, China; 18280446697@163.com (S.W.); 13547802519@163.com (Q.Z.); 2College of Movie and Media, Sichuan Normal University, Chengdu 610064, China; honglexi@126.com

**Keywords:** rumor refutation, disaster-related, NLP, machine learning, XGBoost, group behavior

## Abstract

This research provides a general methodology for distinguishing disaster-related anti-rumor spreaders from a non-ignorant population base, with strong connections in their social circle. Several important influencing factors are examined and illustrated. User information from the most recent posted microblog content of 3793 Sina Weibo users was collected. Natural language processing (NLP) was used for the sentiment and short text similarity analyses, and four machine learning techniques, i.e., logistic regression (LR), support vector machines (SVM), random forest (RF), and extreme gradient boosting (XGBoost) were compared on different rumor refuting microblogs; after which a valid and robust distinguishing XGBoost model was trained and validated to predict who would retweet disaster-related rumor refuting microblogs. Compared with traditional prediction variables that only access user information, the similarity and sentiment analyses of the most recent user microblog contents were found to significantly improve prediction precision and robustness. The number of user microblogs also proved to be a valuable reference for all samples during the prediction process. This prediction methodology could be possibly more useful for WeChat or Facebook as these have relatively stable closed-loop communication channels, which means that rumors are more likely to be refuted by acquaintances. Therefore, the methodology is going to be further optimized and validated on WeChat-like channels in the future. The novel rumor refuting approach presented in this research harnessed NLP for the user microblog content analysis and then used the analysis results of NLP as additional prediction variables to identify the anti-rumor spreaders. Therefore, compared to previous studies, this study presents a new and effective decision support for rumor countermeasures.

## 1. Introduction

Because of the widespread ownership of smart phones and mobile devices, disasters are no longer regional events as anybody can immediately access disaster reports through social media feeds such as Twitter, Facebook or Sina Weibo. It means that human observations from ground zero (actual site of the disaster) can be easily spread, which could be of assistance to attracting needed assistance or optimizing aid programs. However, sometimes, these disaster reports are untrue, exaggerated, or distorted as people not at the disaster site may propagate the information posted by others or cite information links from different sources without bothering to verify the actual facts [1]. In particular, most people are able to access information from the thousands of information channels on Online Social Networks (OSNs) [2]. Vosoughi et al. [3] analyzed all of Twitter’s controversial news stories since its inception; 126,000 retweets by 3 million users; and found that the truth was no match compared to the number of rumors and outright lies as the fake news. Moreover, rumors were found to reach more people, and were able to penetrate deeper and faster on social networks than accurate facts [3]. It was concluded that while there may have been some AI involvement in the misinformation spread, most could be attributed to deep-rooted human nature: that is, people tend to read tweets that confirm their existing attitudes (selective exposure), regard information that is more consistent with their pre-existing beliefs as more persuasive (confirmation bias), and prefer entertaining and incredible content (desirability bias). Besides, group influences have also been found to be related to the factors associated with the social network users (tweet authors, information spreaders, etc.) [4,5]. The users’ retweet intentions were affected by others’ forwarding behavior in groups of close friends, and the users’ retweet likelihood was positively associated with the number of the acquaintances who had retweeted the microblog [5].

While misinformation about celebrities can be dismissed as entertainment, the propagation of misinformation and gossip during a disaster (or concoctive disasters) can cause large-scale panic or even economic losses [6]. However, some people seek to confirm the claims and then spread the confirmations to refute the rumors [7]. Although there are far fewer people refuting than spreading the fake news and rumors [8], there is a probability that the rumors can be refuted or corrected through social media [9]. The timely and accurate refutation against disaster-related rumors is crucial for the maintenance of the order of network environment and public security. Therefore, the goal of the study in this paper is to determine whether these anti-rumor spreaders can be identified through temporal, structural, linguistic, and social tie features [10,11]. As the target group is relatively small, it is valuable to detect these people based on their characteristics and recommend rumor refuting microblogs to them, as once these people receive this information they would be more likely to continue spreading the rumor refuting microblogs than general people and accelerate the social media rumor adjustment and correction process [12,13].

Due to the potential risks of rumor propagation, it is necessary to identify the strategies that can restrain internet rumors. The most identified countermeasures have included two tactics; blocking the rumors to inhibit the spread, and spreading the truth [12]. However, transforming users to become anti-rumor spreaders in OSNs has been found to require significant incentives, and combating rumors needs social support to remove the internet rumors, and could contravene human rights [14]. Two common anti-rumor schemes, random immunization and targeted immunization, could also be improved. Immunization strategies, which is when some internet nodes know the truth, have frequently been used to fight rumors [13]. However, as OSNs has a high degree of redundancy and heterogeneity, random immunization is not suitable as it only works in well-formed homogeneous networks [15]. Targeted immunization, which has been found to be effective in scale-free networks, was developed to effectively quash rumors by seeking out the most highly connected individuals [16]. However, the common approaches to conquering rumors by revealing the truth to users have been relatively ineffective, primarily because of their failure to convince users to trust the veracity of the anti-rumors. Further, the focus on identifying influential nodes or harnessing the power of opinion leaders to refute rumors has neglected to acknowledge the significance of users who are willing to retweet rumor refutations without the need to be persuaded with extra incentives. Therefore, this paper employed targeted immunization to spread the truth, whereby the targeted users voluntarily convince their followers and proactively communicate with irrational users. This research is driven by the belief that if this specific group of people can be identified and taken advantage of, it may be possible to break through the former patterns and gain new insights into more effective countermeasures for internet rumors.

Previous researchers have tended to focus on the rumor spreading process. Among the simulation models, the SIR model [17] was the most famous one. At the very beginning of the rumor propagation process, all individuals are susceptible (S), and as the rumors spread, some become infectious (I) and after the rumor has run its course, all people become recovered (R). Based on the classic SIR model, researchers have extended [2,18] or adjusted [19,20] the original SIR model. In addition, other models have also been developed, such as the Galam Model [21], which uses majority rule reaction-diffusion dynamics, and the Energy Model [22], which is based on heat energy calculations. However, previous works focused on rumor spreading process, and hardly proposed interventions to stop rumors spreading. To the authors’ best knowledge, there has not any research on rumor refutation from an individual level.

Recently, with the rapid development of computer science and technology, a variety of machine learning methods such as logistic regression (LR) [23] and random forest (RF) [24] have been applied in multiple areas. The methods have also been applied in rumor feature detection. Zhang et al. [25] used LR to investigate the important features to distinguish the authenticity of rumors, and Wu et al. [26] used support vector machines (SVM) and Vijeev et al. [27] used SVM and RF for rumor detection. However, to date, no research has been conducted that has applied these algorithms to anti-rumor spreader feature detection. Therefore, this could be a totally different angle from which to approach rumor refuting analyses.

The aim of this research is to construct a model from disaster-related rumor refuting microblogs so that the model can learn the spreader features and detect potential new spreaders. Further, given the information about the viewers of rumor-refuting microblogs of the same major class, the trained model could be used to detect the possible spreaders of a new disaster-related rumor refuting microblog. To realize these aims, we propose a classification model assistant with Nature Language Process (NLP) used for text analysis for crowd identification. In consideration of data amount, it is suitable for machine learning but insufficient for deep learning, and the affiliated goal is to figure out the important classification features. Four types of machine learning algorithms generally used for classification problems, i.e., LR, SVM, RF and extreme gradient boosting (XGBoost), are compared on different rumor refuting microblogs to determine the most efficient detection algorithm, and different feature groups are also compared to validate whether the group behavior information can assist in detecting the target people. New insights into more effective countermeasures for internet rumors are also obtained.

## 2. Materials and Methods

### 2.1. Research Question

Based on previous research, this study seeks to address the following questions:RQ1. How can anti-rumor spreaders be more efficiently identified?RQ2. Is it possible to generalize a trained model for other microblogs if they are of the same class?RQ3. Can group behavior variables help in distinguishing the spreaders?RQ4. What is the validity period of the developed model?

First, to answer these questions and to determine the most efficient model, two comparison studies are conducted between (1) the traditional variables and the traditional variables plus the variables identified in the text analysis, and (2) four machine learning algorithms. 

Second, the possibility of generalizing the spreader detection model is examined. With a focus on RQ1, the model from one particular microblog was trained and then used to predict the potential spreaders in another microblog to assess whether the trained model could be used for other microblogs in the same class (disaster-related). 

Third, as different microblogs vary in terms of how often they are retweeted, commented on, or liked, if a microblog has been widely retweeted, does this encourage further dispersion? To explore this question, three group behavior variables are included in the original model. 

Finally, because many users regularly update their microblogs, the most recent (updated in the previous week) microblogs were collected to ensure a focus on the users’ most recent interests, with the content being taken as the prediction variables. However, as RQ4 is focused on model timeliness, are microblogs from a month ago helpful in improving the prediction precision? The overall research framework is shown in Figure 1.

### 2.2. Data Collection

Sina Weibo is a Chinese microblogging (Weibo) website, and is one of the most popular social media platforms in China with 431 million active monthly users. Unlike WeChat, which only allows a user to post to certified friends, Sina Weibo is more similar to Twitter as it has a wider, more open dispersal. Therefore, because of this feature, crawling the microblogs on Sina Weibo has high research value for rumor spreading or rumor refutation spread analyses. Using a web crawler, rumor-refuting microblogs related to four natural or man-made disasters were extracted (two related to earthquakes and two related to a conflagration/explosion; some of which were official declarations and some of which were not:Refutation of earthquake forecasting in Fujian, China from the “China Seismological Station Rapid Report” (2018-11-26; 395 Retweets, 2050 Comments, 2003 Likes).Refutation of the false video in Yibin, China after a magnitude 5.7 earthquake from “Sichuan Releasing” (2018-12-16; 1162 Retweets, 964 Comments, 2360 Likes).Refutation of the false explosion video in Queens, New York by “Weibo Refutes Rumors” (2018-12-19; 242 Retweets, 146 Comments, 809 Likes) and “Capture Rumors” (2018-12-19; 362 Retweets, 112 Comments, 144 Likes).Refutation of the conflagration at the Tencent Building in Beijing by “Tencent News” (2018-12-21; 7092 Retweets, 3592 Comments, 5201 Likes).

For these four events, 38365 microblogs from 3999 Weibo users (3793 valid) were extracted. Except for few users who only tweet less than 11 microblogs, 11 microblogs were extracted. Extracted information included the topping microblog. Although the topping microblogs had not been recently posted, they were able to reveal the overall attitude of the user to some extent. Basic user information; gender, membership level, location, birthday, verified information, signature or brief introduction, label, logging access in Sina Weibo, microblog number, number of followers and following, and group numbers for each user were also extracted.

As it is not possible on Sina Weibo to know how many people viewed a particular microblog, information was mined from the comment lists and the retweet lists as it was assumed that the people who commented or retweeted must have already viewed the microblog. The like list, however, was ignored as the aim of the research was to identify the number of people who only commented (stiflers) and the number of people who retweeted and commented on or only retweeted (spreaders) to ensure that there were a wide range of attitudes in the training data set. 

User information was collected based on the retweet/comment time and this data was used to train a model to predict the behavior of others. Three to five times more comments were collected than retweets for the different rumor refutation news. Obviously, because of the selective data collection process, there were far more stiflers and spreaders than there would be in reality as generally, only 2–3 out of 100 would be potential spreaders. However, as it is difficult to compare algorithmic efficiency if the ratio is set that high without having to crawl millions of data, simply testing the various algorithms on a small subset of data [28] was considered powerful enough to select the appropriate algorithm, after which the most robust and efficient machine learning model could be determined through simulation. 

The top 20% of spreaders were selected as the target group, with the other users formed the training data set. The test data set was then developed to ensure that there was the same ratio of stiflers and spreaders as in the training data set. As the goal was to distinguish the target group, the test data set was a combination of the target group and the users who were not in the retweet/comment/like list of the four research microblogs. 

We originally introduce the results of text analysis of each individual as the prediction factors of our classification model. To convert the texts into measurable numerical variables, Baidu’s AipNLP was applied, which is regarded as the most advanced Chinese text analysis technique. The goal of sentiment analysis is to convert the text into a value between 0 and 1; the closer the value is to 1, the more positive the text sentiment, and the closer the value is to 0, the more negative the text sentiment. Similar to sentiment analysis, short text similarity analysis converts a text into a value between 0 and 1 given another matched text, and the higher the value, the more similar the two texts. To ensure precision, the data were cleaned using the following process: Rumor refuting microblogs were deleted if they were one of the 11 most recent microblogs from the user.A sentiment value of 0.5 and a similarity value of 0.5 were assigned to the microblogs to verify the information or signatures if the text was missing.Words irrelevant to the content of the text but would significantly influence the result of sentiment analysis, such as “Like”, “Collect”, “Honor” (a mobile phone brand), were removed.

The final research variables are shown in Table 1.

### 2.3. Prediction Based on Machine Learning Methods

#### 2.3.1. Brief Overview of the Machine Learning Methods employed 

In this paper, to answer RQ1, the following four machine learning methods were used.

Logistic regression (LR) is a famous statistical model based on a corresponding probability of which class the dependent variable belongs to. Mathematically, a binary logistic model has a dependent variable with two possible values, such as win/lose, and in this research repost/not repost. 

Support vector machines (SVM) [29] is a binary classifier, and it tries to minimize an upper bound of the generalization error [30]. Due to the low requirement on training samples, the SVM has good generalization performance regarding small samples [31,32]. However, it directly assigns new examples to one category based on the marked training data. 

Random forest (RF), which uses multiple decision trees, has also been frequently used for classification. In this model each decision tree is a classifier, and the RF algorithm combines all classification results and makes the final classification decision using voting. RF has proved to be a very precise and robust algorithm [33].

XGBoost [34] is a newly developed algorithm that has been gaining popularity in industrial and data analysis circles. Unlike RF, it uses another kind of tree named the CART (Classification and Regression Tree) and trains the trees serially and interactionally rather than in parallel and independently [34]. 

In order to compare the efficiency for four methods, we applied the same factors for all four models. Python Sklearn and XGboost packages were used.

#### 2.3.2. Prediction Including Text Analysis

Traditional spreader prediction methods have usually used the users’ basic information such as gender, location, and the followers’ scale as the variables, as shown in Table 2. The prediction results of the traditional variables were compared with the variables in Table 1.

To compare the prediction results, three indexes were introduced; Precision, Recall, and the F1 Score. In classification problems, as shown in Table 3, a confusion matrix with four indexes, i.e., TP (True Positive), FN (False Negative), FP (False Positive), TN (True Negative) was applied. 

Accuracy, which uses the number of correctly predicted samples divided by the total sample size as a ratio, is obviously not available for classification problems. If the category quantity is imbalanced; for example, 90 class 0 and 10 class 1; the classifier can easily predict that all samples are class 0 and achieve an accuracy ratio of 90%; however, this is misleading as category imbalances are quite common in real world problems.

Therefore, to overcome this type of inaccuracy, Precision, Recall and F1 Score were introduced, the calculations for which are as follows [35]:(1)$$\mathrm{Precision}=\frac{\mathrm{TP}}{\mathrm{TP}+\mathrm{FP}}$$
(2)$$\mathrm{Recall}=\frac{\mathrm{TP}}{\mathrm{TP}+\mathrm{FN}}$$
(3)$$F1\;\mathrm{Score}=\frac{2*\mathrm{Precision}*\mathrm{Recall}}{\mathrm{Precision}+\mathrm{Recall}}$$

We can interpret Precision as the probability that a (randomly selected) retrieved document is relevant, and Recall as the probability that a (randomly selected) relevant document is retrieved in a search. F1 Score combines Precision and Recall as a harmonic mean of the two, so F1 Score does not suffer when the class sizes are imbalanced. We used macro-averaged Precision, Recall and F1 Score which were average values of each class [36]. The Precision, Recall and F1 Score were all based on the result of the testing data set.

The results for the control group are shown in Table 4.

As shown in Table 4, using the traditional variables when the text analysis was not included, the prediction results were very unstable. While there were reasonable values from the F1 Score for the Random Forest and XGBoost Models (Microblog1 and Microblog4), the prediction results were also poor (Microblog2 and Microblog3). Therefore, the results can be poor if the prediction power strongly depends on the microblog itself. The SVM and LR F1 Scores were mostly around 0.5, except for Microblog1 with LR and Microblog4 with SVM.

However, in the new model in which the text analysis was included, the prediction power was more robust. The SVM and LR models produced a relatively stable F1 Score of 0.6–0.7, and the RF and XGBoost models produced a stable F1 Score of 0.7–0.8. Comparing the results of XGBoost and RF, we found that the average F1 Score for XGBoost was 0.747, and the average F1 Score for RF was 0.709. Simultaneously, the standard deviation of XGBoost was 0.0303, which was smaller than that of RF (0.0384). The XGBoost model was relatively more efficient among the four models, with a constant F1 Score of around 0.75 for any of microblog. The robustness and prediction power both substantially increased using the proposed model, indicating that the inclusion of the sentiment and similarity text analysis significantly improved the prediction power.

Generally, to predict whether a Sina Weibo user was a potential disaster-related anti-rumor spreader, the XGBoost had the best performance, followed by RF, with the SVM being slightly better than the LR.

By applying XGBoost model, it is possible to identify the feature importance for the prediction, as shown as Figure 2. When XGBoost is constructing a tree, it needs to split to make sequential classifications, and the feature importance score is judging the information gain of different features. The more information gains a feature can provide, the more likely it would be selected by XGBoost as the basis of classification, and the higher of the feature importance score. Although the feature importance characteristics were different for the different microblogs, there were still some noticeable similarities. For instance, the ‘Number of Microblogs’ was always an important feature as it was a good measure of a user’s activity, while the Number of Users a user followed, Gender, Membership Level and Location were not as important as expected. On the contrary, the content of the users’ verified information, the signature or brief introduction and the 11 most recent posted microblogs (including the topping microblog if they had one) were found to play an important role in spreader prediction.

#### 2.3.3. Generalization Ability of Our Model

This section answers RQ2 to check the generalizability of the model. Considering the scenario where we have trained a model to predict spreaders, now we observe a new rumor-refuting microblog and wish to find potential spreaders to recommend this microblog to. Is it possible to use the trained model for distinguishing even if the training set’s microblog topics are not exactly the same as the new rumor-refuting microblogs?

To assess these suppositions, heuristics were performed. First, datasets from the different microblogs but under the same earthquake subclass (Microblog1 and Microblog2) and the same fire/explosion subclass (Microblog3 and Microblog4) were combined. Then all the data were combined as a major disaster class data set. The combined data set was then used for the training and testing, and the F1 Score was used to evaluate the prediction results.

As it is shown in Table 5, after combining the data sets, there was a slight increase observed in the prediction power, and the fluctuation of classification results also decreased because of the additional data, which tended to indicate that when the data was aggregated under the same subclass or major class, given enough data, the XGBoost Model and model variables could be used to judge whether a new given user was/was not a principal spreader.

After combining the data sets, there was a slight increase observed in the prediction power, and the robustness also increased because of the additional data, which indicated that when the data was aggregated under the same subclass or major class, given enough data, the XGBoost Model and model variables can be used to judge whether a new given user was/was not a principal spreader.

Next, the trained model based on the earthquake subclass was used to predict the spreaders in the fire/explosion subclass, and then the trained model based on fire/explosion subclass was to predict the spreaders in the earthquake subclass, the results for which are shown in Table 6.

As can be seen, the prediction powers all decreased, clearly indicating that the earthquake and fire/explosion belonged to different subclasses. However, as the features were only learned from two sample microblogs before the trained model was used to predict another two microblogs, there was possibly not enough data for the trained model to be able to develop the skills for correct feature extraction and machine learning. However, the RF and XGBoost models both had reasonable predictions with F1 Scores between 0.6–0.65.

#### 2.3.4. Group Behavior on Prediction

This section is to answer RQ3, namely to work out whether the public would be affected by conformity, matching their retweet behaviors to group norms and conforming to the majority of people’s actions. Three additional variables, as shown in Table 7, are introduced to reflect group reactions, i.e., PAR, PAC, PAL, in addition to the original variables. Generally, by comparing one microblog post’s PAR, PAC, PAL with others, users could detect anti-rumors’ popularity (e.g., the bigger the PAR, the more popular that rumor refuting microblog is) and speculate on groups’ attitudes (e.g., the bigger the PAL, the more users like this post). This section validates whether group behavior influenced the microblog retweet behavior and if users acknowledged the number of people who have already retweeted, commented on or liked the current microblog. 

As the user information regarding retweet/comment times has already been crawled as part of the initial user information, it is relatively easy to extract the number of people who have previously retweeted or commented. The two other indexes, PAC+PAL/PAR+PAL were then surmised based on the proportion between the total number of people who retweeted, commented on or liked.

As the data indicated that there were far less spreaders than stiflers, with spreaders only being 1/10 of the total retweets and stiflers being responsible for nearly 1/2 of the total comments (for a microblog, the number of retweets and comments are similar), the spreaders would tended to have higher PAR, PAC and PAL, which would result in an unrealistically high F1 Score. Therefore, the data needed to be adjusted to ensure that the spreaders and stiflers for each microblog had the same scale and same proportion for the number of retweets/comments. By employing a new dataset, we tried to let the PAL of spreader and stifler become equivalent in value range.

As different microblogs have different popularity, it is meaningless to consider the PAR, PAC and PAL for a single microblog especially after ensuring that the scales are adjusted for each microblog. Therefore, the data sets from the different microblogs were combined to form a new data set that include information from 2223 valid users; 445 data of which were used for testing, 234 of which were stiflers and 211 of which were spreaders. Therefore, the category imbalance problem was solved by changing the dataset. The goal of this analysis, however, was not prediction; rather, the goal was to explore whether group behavior had any role in the possibility of predicting spreaders. 

As can be seen in Table 8, there was little difference between the F1 Scores when group behavior was/was not considered, which indicated that considering group behavior did not improve the prediction of spreaders when added to the original set of features. However, Figure 3 indicated that PAC, PAR and PAL had an important role in the prediction, from which it can be concluded that group behavior was shown to have some correlation with individual behavior but did not contribute to identifying possible spreaders.

#### 2.3.5. Timeliness of the Model

This section is to answer RQ4, regarding timeliness of the model. For an application-oriented model, timeliness is crucial. Therefore, the data was collected the moment the retweet and comments become stable so that the most recent microblogs were automatically collected. As demonstrated in Part 3, most microblogs were updated in the previous week. As it has been proven that applying text analysis to disaster-related anti-rumor spreader detection was helpful, the question remained as to whether the text analysis would still be valid if microblogs that had been updated one month previously were used?

As it is technically hard to get the user microblogs one month before the rumor refuting microblog, we used the microblogs one month after it as a substitute. To some extent, they are equivalent for our research purpose.

Using the same method and parameters, the F1 Scores for each data set and each method were calculated and compared. As can be seen in Table 9, even though the prediction results can be considered reasonable (Microblog1 and Microblog4), the prediction power was generally lower, and the F1 Scores fluctuated. For example, the XGBoost returned a good result with an F1 Score of 0.726 (Microblog1) and a poor result with an F1 Score of 0.524 (Microblog2), and with the F1 Scores for Microblog2 and Microblog3 being relatively weak. Therefore, as the proposed prediction method was shown to be strongly time-based, the most recent user microblogs would need to be used for the text analysis data source.

## 3. Results

First, to predict the anti-rumor spreaders, it was found that the sentiment and similarity text analysis was important, and that the XGBoost model was able to achieve the most optimal performance. The strong correlation was tested under machine learning models using a controlled experiment between the groups with and without text analysis, from which it was found that the former group was more stable. It indicated the inclusion of text analysis improved model performance. When the XGBoost model was used to calculate and rank the influence of the user features on prediction, even though the same outcomes were not achieved for all microblogs, it was found that the number of user microblog posts was constantly ranked highly. Moreover, the users’ verified messages, signatures or brief introductions, and the 11 most recently posted microblogs (topping microblogs are included) were reliable predictors, while the number of users a subscriber follows, gender, location, and membership level were not reliable predictors. What makes these findings distinctive from prior studies is that many of Sina Weibo’s peculiarities such as verified information and the signature are considered and proven to be crucial to the prediction analysis.

Second, the trained model’s predictive ability was shown to be relatively stable when adopted in other micro posts of the same class. However, the model’s predictive power was not as effective when used on a different subclass such as using earthquakes to test fire/explosion. However, given a larger sample size, we argue it is possible that these predictors could be used to identify anti-rumor spreaders as long as the provided data come from the same major class.

Third, interestingly, while group behavior was observed to be relevant to users’ retweet actions, it was less helpful for spreader recognition. Even though three additional variables; PAR, PAC and PAL; were included, which respectively represented the number of other users who have previously retweeted/commented on/liked the posts, there was little increase in the F1 score.

Fourth, the experimental results demonstrated that time was a determining factor in the prediction models, indicating that the users’ latest micro posts needed to be collected for the text analysis because the bloggers’ most recent concerns were strongly associated with their real-time tweets. When bloggers focused on different things in different time periods, their post content might vary widely. The unstable and incoherent F1 scores when the microblog time durations were varied indicated that the proposed prediction method was time based.

## 4. Discussion

The rise of social media platforms has allowed people to communicate in real time making it easier for everybody to be a communicator. Therefore, while microblogs have allowed people to more easily exchange information and share opinions, they have also allowed for misleading information and rumors to be widely distributed, which can cause harm to public security and society.

Many methods and algorithms have been developed for tracking and predicting rumor propagation. However, to be able to identify erroneous messages requires a combination of computer text analysis and human judgment. Further, even though the truth is available, many bloggers cannot be persuaded to accept the facts because of their inherent ideologies and personal preferences.

This paper sheds new light on rumor refutation as the purpose of the current study is to recognize and predict the types of individuals that are most likely to retweet anti-rumors. Opinion leaders guide ideas, agitate the public to understand social problems and form and correct public opinion [37]. Unlike traditional opinion leaders with enormous influence, microblog opinion leaders are generally unknown but are connected to many groups of people. Research has shown that people are more inclined to trust information from their own social groups than they are from the news media [38]. However, it only takes one person in this group to clarify information and refute the rumor. Therefore, public participation in rumor-refutation could comprehensively reduce the effects of rumors and also provide sufficient institutional guarantees for the government.

The findings in this study provide us with guidance on restraining rumor propagation. First, by accurately delivering anti-rumors, the predicted opinion leaders would be more likely to notice them and more motivated to disseminate them, which could gradually change others’ attitudes towards misinformation. Second, after identifying the anti-rumor spreaders, they could be subdivided even further based on their followers’ characteristics, so more tailored contents and more persuasive factual evidence could be sent to different groups.

This research is to identify the people in a given group who would be more likely to forward anti-rumors, but is limited by the choice of only four disaster events. Nonetheless, the findings in this research have given rise to possible future research avenues. For example, it would be interesting to examine whether these methods would be viable on other platforms such as WeChat or Facebook. As WeChat works on stable closed-loop communication channels in private space, such private social networks could highlight the importance of groupthink. Another area of interest would be to examine whether the model holds true for different types of rumors; therefore, future studies could examine diverse types of rumors and examine whether it is also possible to identify the possible anti-rumor users.

## 5. Conclusions

This aim of this study was to develop a machine learning method that could identify the users who would be more likely to refute rumors, from which the following conclusions were drawn. (1) The addition of text analysis was found to fundamentally improve the traditional prediction model. (2) The trained model was found to have good generalization ability and could be widely applied as long as the original and applied microblogs belong to the same class. (3) Group behavior variables were not significant in predicting possible retweeters. (4) The timing was found to significantly affect model validity and therefore choosing the “most recent” blogs was necessary for accurate analyses.

Jonathan Swift once said: “Falsehood flies and truth comes limping after; so, when men come to be undeceived, it is too late” [39]. Dealing with rumors has never been an easy task. The present study was designed to predict the people who would step forward and refute the rumors. However, identifying these key opinion leaders is only the first step towards controlling online rumor propagation. Therefore, future studies need to develop more accurate prediction and recommendation mechanisms based on empirical research to effectively combat online rumor propagation. Choosing more disaster events and larger sample sizes are also needed to illustrate the generality of our results. More broadly, the study could extend into various social media platforms to enhance applicability of our model and optimize the method.

## Figures and Tables

**Figure 1 ijerph-16-01452-f001:**
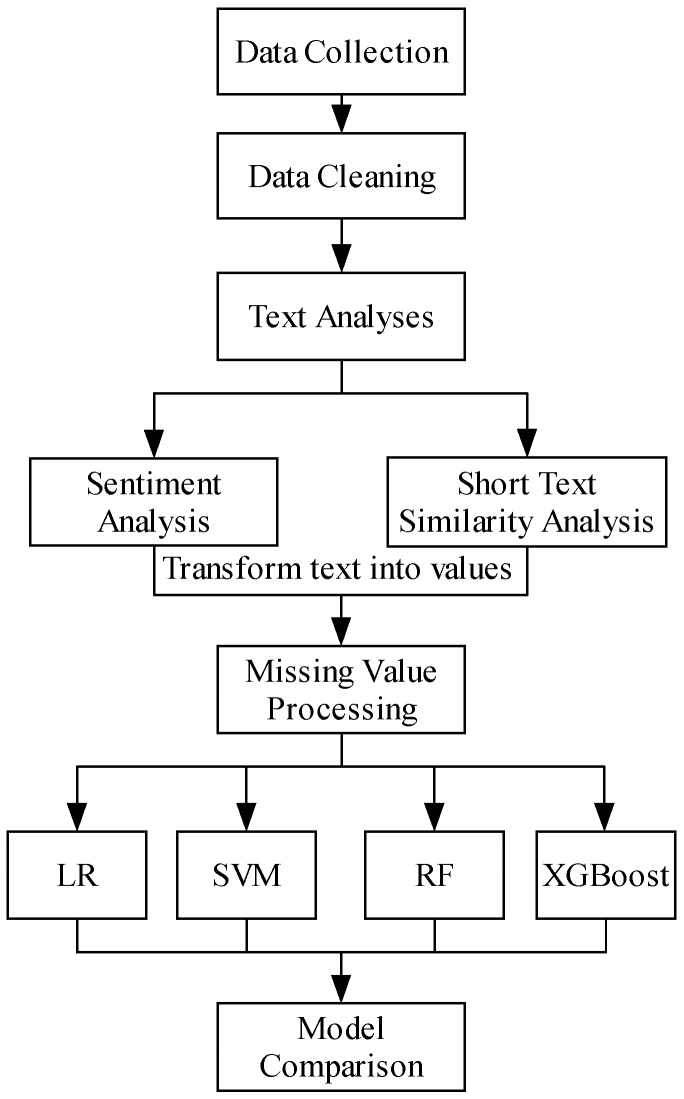
Research methodology framework.

**Figure 2 ijerph-16-01452-f002:**
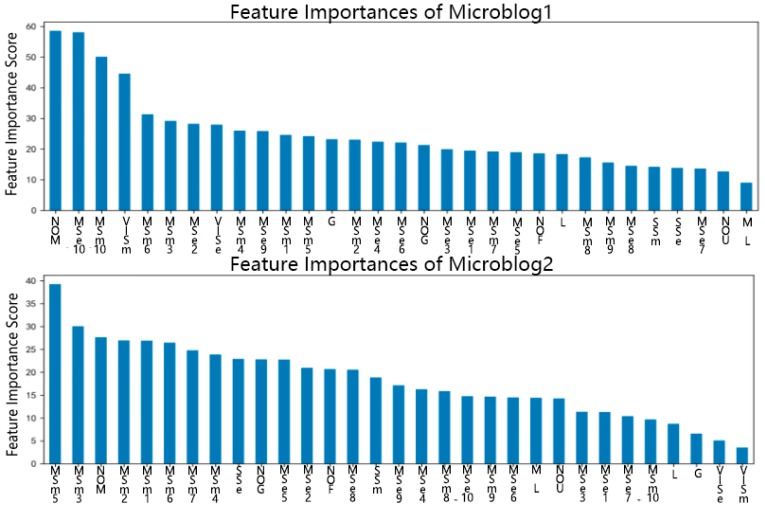
Feature importance bar plot based on XGBoost.

**Figure 3 ijerph-16-01452-f003:**
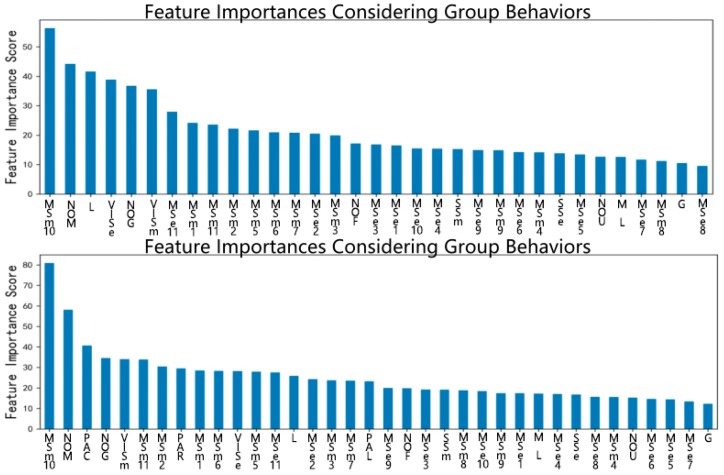
Feature importance based on XGBoost.

**Table 1 ijerph-16-01452-t001:** Research Variables.

Variable	Description
R	The dependent variable. Whether the user retweeted the rumor refuting microblog. 1 for yes and 0 for no.
G	The gender of the user set. 1 for male and 0 for female.
ML	Membership Level. Explains the users’ devotion and activity to some extent.
L	Location. The provincial level location of the user. 1 if the user was in the same province as the rumor, 0 otherwise.
VISe	Sentiment of the verified user information ranging from 0–1; the larger the value, the more positive the attitude.
VISm	Similarity between the verified information and the rumor refuting microblog ranging from 0–1; the larger the value, the more similar the two texts.
SSe	Sentiment of the signature or brief introduction of the user ranging from 0–1; the larger the value, the more positive the attitude.
SSm	Similarity between the Signature or Brief Introduction of the user and the rumor refuting microblog ranging from 0–1; the larger the value, the more similar the two texts.
NOM	Number of Microblogs the user has already posted.
NOU	Number of Users the user has followed.
NOF	Number of Followers the user has.
NOG	Number of Groups the user has classified as friends.
MSe (1–11)	Sentiment of the *i*th microblog of the user ranging from 0–1; the larger the value, the more positive the attitude.
MSm (1–11)	Similarity between the *i*th microblog of the user and the rumor refuting microblog ranging from 0–1; the larger the value, the more similar the two texts.

**Table 2 ijerph-16-01452-t002:** Variables for the control group.

Variables	Description
R	The dependent variable. Whether the user retweeted the rumor refuting microblog. 1 for yes and 0 for no.
G	The gender of the user set. 1 for male and 0 for female.
ML	Membership Level. Explains the users’ devotion and activity to some extent.
L	Location. The provincial level location of the user. 1 if the user was in the same province as the rumor; 0 otherwise.
NOM	Number of Microblogs the user has already posted.
NOU	Number of Users the user has followed.
NOF	Number of Followers the user has.
NOG	Number of Groups the user has classified as friends.

**Table 3 ijerph-16-01452-t003:** Binary confusion matrix.

Confusion Matrix		Predict	
		0	1
Real	0	TP	FN
	1	FP	TN

TP: Valued 0 and predicted to be 0. FN: Valued 0 but predicted to be 1. FP: Valued 1 but predicted to be 0. TN: Valued 1 and predicted to be 1.

**Table 4 ijerph-16-01452-t004:** F1 Scores of control group results for the four machine learning models.

Microblog	Control Group	SVM	LR	RF	XGBoost
Microblog1	Traditional	0.521	0.631	0.721	0.676
	New	0.634	0.605	0.743	0.721
Microblog2	Traditional	0.528	0.456	0.53	0.605
	New	0.67	0.637	0.675	0.791
Microblog3	Traditional	0.536	0.587	0.413	0.592
	New	0.642	0.624	0.677	0.741
Microblog4	Traditional	0.693	0.593	0.716	0.627
	New	0.639	0.639	0.742	0.736

**Table 5 ijerph-16-01452-t005:** F1 scores of subclass and major class predictions.

Data Set	SVM	LR	RF	XGBoost
Earthquake	0.652	0.652	0.741	0.772
Fire/Explosion	0.653	0.67	0.741	0.767
Major Disaster Class	0.636	0.658	0.714	0.778

**Table 6 ijerph-16-01452-t006:** F1 scores of cross-subclass prediction.

Methods	SVM	LR	RF	XGBoost
Earthquake Predicts Fire/Explosion	0.591	0.466	0.627	0.618
Fire/Explosion Predicts Earthquake	0.519	0.600	0.611	0.631

**Table 7 ijerph-16-01452-t007:** The three introduced group behavior variables.

Variable	Description
PAR	The number of other people who have retweeted the rumor refuting microblog prior to the user doing so.
PAC	The number of other people who have commented on the rumor refuting microblog prior to the user doing so.
PAL	The number of other people who have liked the rumor refuting microblog prior to the user doing so.

**Table 8 ijerph-16-01452-t008:** F1 scores of considering and not considering group behavior.

Methods	SVM	LR	RF	XGBoost
Not Considering Group Behavior	0.738	0.748	0.789	0.827
Considering Group Behavior	0.75	0.776	0.795	0.816

**Table 9 ijerph-16-01452-t009:** The F1 Score considering duration after rumor refuting.

Microblog	Control Group	SVM	LR	RF	XGBoost
Microblog1	Immediately	0.634	0.605	0.743	0.721
	After 1 Month	0.597	0.64	0.721	0.726
Microblog2	Immediately	0.67	0.637	0.675	0.791
	After 1 Month	0.568	0.467	0.618	0.524
Microblog3	Immediately	0.642	0.624	0.677	0.741
	After 1 Month	0.579	0.593	0.626	0.621
Microblog4	Immediately	0.639	0.639	0.742	0.736
	After 1 Month	0.646	0.605	0.702	0.669

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
