# Peer review of "Machine Learning Methods to Predict Social Media Disaster Rumor Refuters"

_ijerph, 2019, doi:10.3390/ijerph16081452_

Round 1
Reviewer 1 Report
The authors propose a method for identifying people who refute rumors about disasters on social media. The method exploits user and microblog data from social networks and sentiment and text similarity analysis to identify who is likely to retweet rumor-refuting microblogs. Similarity and sentiment are found to boost performance significantly.
The task is well motivated (countering rumours by spreading the truth -- need to identify the true refutators because other strategies have practical problems). The paper is well-structured and the task and method descriptions in section 2 are quite detailed. However, the experiments would benefit from a clearer description -- please see the comments listed by line below. A larger problem was that I could not follow the experiment description for the group behavior variables. Also, for the first experiment, there should be some more discussion of the reasons for differences between random forest and XGBoost methods -- we need to know something about why XGBoost performs better if we are to believe this improvement will hold on new disaster topics.
The experiments are also limited by the choice of only four disaster events. This makes it difficult to draw strong conclusions about whether the model generalizes from one subclass to another. This limitation should be noted in the paper. However, I think the results are nonetheless a good contribution.
Comments by line (mostly minor comments):
Line 88-97: you mention three models -- do these previous works simply model the rumour spreading process, or do they propose interventions to stop rumours spreading?
Line 100: the general examples you cite seem unnecessary and too unrelated. I would cut this line and provide the citations for the specific machine learning methods and more related tasks, such as rumour detection.
Line 108: ‘microblogs’
Line 109: this sentence is unclear. Does ‘given the viewer information from the same major class’ mean ‘given information about the viewers of rumor-refuting microblogs of the same class’?
Line 144: ‘rumours … were extracted’ -- should this really say ‘rumour-refuting microblogs … were extracted’?
Line 157: please clarify: “users who did not have sufficient microblogs” means users who did not have sufficient microblogs related to the four disasters?
Line 207: and SVM is not necessarily linear, unless you use a linear kernel.
I don’t know if it is possible to make the general claim about SVM performance on lines 208-209. Can you cite some examples?
Lines 213-214: similar to the previous comment, can you give more specific examples about when an RF outperforms other classifiers? I am not sure it is true to say it is always more robust and precise than other classifiers.
Line 217: what is the benefit of the XGBoost approach over RF? When did it outperform RF and why?
Line 236: this is not quite right. Accuracy is available for binary classification, but it can be misleading, as you state. It is also better to say it is misleading than it is not accurate.
lIne 245: rather than simply state that F1 score is a good measurement, say that it does not suffer when the class sizes are imbalanced
Line 262: does ‘significantly’ here mean ‘statistically significantly’? It would be good to include a test for statistical significance here. If you do not, please use another word to avoid confusion, such as ‘substantially’.
Figure 1: what is the feature importance score? I understand this is specific to XGBoost, but it should have a brief description since it is not commonly known.
Throughout section 2, please refer back to the RQs in section 2.1 to clarify which set of results corresponds to each research question.
Line 282: this paragraph is very unclear. What do ‘above prediction process’, ‘under the same microblog’ and ‘current information’ refer to? I think you want to say something like “Consider the scenario where we have trained a model to predict spreaders. Now we observe a new rumor-refuting microblog and wish to find potential spreaders to recommend this microblog to. Is it possible to distinguish whether a user who was not in the training set will be a spreader for this microblog?”
Line 296: what do you mean by robustness here, and how are you testing it? I see only F1 scores in the table, which do not vary much between the three subclass/major class setups.
Line 300-304: some of the text from the previous paragraph is repeated.
Line 310-315: “the features were only learned from two sample microblogs before the trained model was used to predict”-- why is the training set different in this case to table 5? It should be trained on the same datasets to be comparable.
Lines 329 to 341: I cannot understand these paragraphs. Why would the PAR, PAC and PAL values be so different for spreaders and stiflers? What is this rescaling step that is needed and why? What does ‘heat’ mean in this context? Why does there need to be a new dataset rather than using the same dataset from 2.3.3 (RQ2)?
Line 348-352: you say that group behavior did not contribute to identifying possible spreaders, but I think this is better phrased as “did not improve prediction of spreaders when added to the original set of features”. There is a correlation, and if you did not have the same set of original features, it is possible that group behavior would improve prediction.
Line 361: After a major disaster, users may still be commenting on the disaster for more than a month. Therefore, it may not be realistic to use data from after the event as a substitute for data from before the event. This might explain the stronger performance on Microblog 1. Perhaps you can comment on this?
Lines 432 - 437: this seems to be just speculation and I don’t think it should be included in a scientific paper unless you provide some supporting evidence. It is quite off-topic as well: the paper is about detecting people who spread rumor-refuting blogs, yet this paragraph talks about the role of government and education. Many people would disagree that the government should interfere in social media, so this is a controversial topic that is probably too complex to mention briefly in passing.
Typos and grammar:
Line 13: natural language processing
Lines 72/73: commas instead of semicolons; should it say ‘anti-rumor schemes’?
Line 126: no semicolon needed
Line 158: should ‘topping’ be ‘topic’ or ‘disaster topic’?
Line 212: full stop/period should be used instead of comma before ‘it is’.
Line 235: should be “which uses the number of correctly predicted samples divided by the total sample size as a ratio,”
Table 9 -- not good to split across pages
Line 384: ‘prediction’ not ‘prevision’?
Line 385: ‘ranked highly’ not ‘ranked top’?
Line 438: ‘Micro-post’ should be ‘micro-post’
Author Response
Response to Reviewer 1
We have read the comments from you very carefully. We are grateful to you for your suggestions in improving this paper. Certainly, it has helped us to clarify several issues and hence, improved the paper.
We then give point-to-point response to the comments in the following, where the comments are marked with «.
«The authors propose a method for identifying people who refute rumors about disasters on social media. The method exploits user and microblog data from social networks a sentiment and text similarity analysis to identify who is likely to retweet rumor-refuting microblogs. Similarity and sentiment are found to boost performance significantly.
« The task is well motivated (countering rumours by spreading the truth -- need to identify the true refutators because other strategies have practical problems). The paper is well-structured and the task and method descriptions in section 2 are quite detailed. However, the experiments would benefit from a clearer description -- please see the comments listed by line below. A larger problem was that I could not follow the experiment description for the group behavior variables. Also, for the first experiment, there should be some more discussion of the reasons for differences between random forest and XGBoost methods -- we need to know something about why XGBoost performs better if we are to believe this improvement will hold on new disaster topics.
[Our Response]: Thanks a lot for your comments and sorry for the unclear description of the group behavior variables. In the revised paper, we described the meaning of PAR,PAC,PAL more detailed and clearer in Table 7.
In addition, we gave more detailed discussion of the reasons for differences between random forest and XGBoost methods, as shown in the line 269-273: “Comparing the results of XGBoost and RF, we found that the average F1 Score for XGBoost was 0.747, and the average F1 Score for RF was 0.709. Simultaneously, the standard deviation of XGBoost was 0.0303, which was smaller than that of RF (0.0384). The XGBoost model was relatively more efficient among the four models, with a constant F1 Score of around 0.75 for any of microblog.”
« The experiments are also limited by the choice of only four disaster events. This makes it difficult to draw strong conclusions about whether the model generalizes from one subclass to another. This limitation should be noted in the paper. However, I think the results are nonetheless a good contribution.
[Our Response]: Thanks a lot for your comments! Yes, we have to admit the drawbacks of only choosing four disaster events. In the revised conclusions, we acknowledged the limitations and promised to increase the sample size in future work, as shown in the line 478-480: “Choosing more disaster events and larger sample size are also needed to illustrate the generality of our results. More broadly, the study could extend into various social media platforms to enhance applicability of our model and optimize the method.”
« Line 88-97: you mention three models -- do these previous works simply model the rumour spreading process, or do they propose interventions to stop rumours spreading?
[Our Response]: Thanks a lot for your comments! These works concentrated on simulation of rumor spreading process, they studied little about interventions to stop rumours spreading. To clarify that, we modified in line 95-96: “However, previous works focused on rumor spreading process, and hardly proposed interventions to stop rumours spreading.”
« Line 100: the general examples you cite seem unnecessary and too unrelated. I would cut this line and provide the citations for the specific machine learning methods and more related tasks, such as rumour detection.
[Our Response]: We have to admit that those examples are unnecessary, so we removed these examples as well as citations, as shown in line 98-100
« Line 108: ‘microblogs’
[Our Response]: Thanks a lot for that! We have already revised it.
« Line 109: this sentence is unclear. Does ‘given the viewer information from the same major class’ mean ‘given information about the viewers of rumor-refuting microblogs of the same class’?
[Our Response]: Thanks a lot for your comments! Its meaning is exactly the same as what you have suggested. We revised it accordingly.
« Line 144: ‘rumours … were extracted’ -- should this really say ‘rumour-refuting microblogs … were extracted’?
[Our Response]: Thanks a lot for your comments! Yes, it is. We revised it accordingly in line 147.
« Line 157: please clarify: “users who did not have sufficient microblogs” means users who did not have sufficient microblogs related to the four disasters?
[Our Response]: Sorry for the unclear statement. It means there are some users who only tweeted several microblogs (e.g., 2 or 3 microblogs) since their registrations and we can just get the amount of microblogs they tweeted. While most users tweet hundreds of microblogs and we only extract the most recent 11 microblogs of them. We clarified it in line 160: “Except for few users who only tweet less than 11 microblogs, 11 microblogs were extracted.”
« Line 207: and SVM is not necessarily linear, unless you use a linear kernel.
I don’t know if it is possible to make the general claim about SVM performance on lines 208-209. Can you cite some examples?
[Our Response]: Thanks a lot for your comments! It is true that SVM is linear only when it applies a linear kernel. We have provided references about the general claim of SVM in line 215-218.
« Lines 213-214: similar to the previous comment, can you give more specific examples about when an RF outperforms other classifiers? I am not sure it is true to say it is always more robust and precise than other classifiers.
[Our Response]: Thanks a lot for your comments! In the literature [Do we Need Hundreds of Classifiers to Solve Real World Classification Problems?---from Manuel Fernandez-Delgado, Eva Cernadas, Senen Barro], authors evaluated 179 classifiers arising from 17 families, they found that the RF is clearly the best family of classifiers regarding the maximum accuracy, therefore, we made the statement. However, when we read your comments, we rethought and decided to revise such statement. It is still not proper to make such a general claim without specific problem settings and evaluation criteria. We revised our statement in line 221-222: “RF has proved to be a very precise and robust algorithm.”
« Line 217: what is the benefit of the XGBoost approach over RF? When did it outperform RF and why?
[Our Response]: Thanks a lot for your comments! Although both XGBoost and RF are tree structured algorithms, they have quite different methodologies. RF trains the decision trees in parallel and makes final decision by voting, while XGBoost trains each new instance to emphasize the training instances previously mis-modeled for better classification result. The idea of XGBoost is innovative while it doesn’t necessarily outperform RF. The performances of XGBoost and RF depend on the data and the assignments of their hyperparameters.
« Line 236: this is not quite right. Accuracy is available for binary classification, but it can be misleading, as you state. It is also better to say it is misleading than it is not accurate.
[Our Response]: Thanks a lot for your comments! We’ve already revised it, as shown in line 247.
« Line 245: rather than simply state that F1 score is a good measurement, say that it does not suffer when the class sizes are imbalanced.
[Our Response]: Thanks a lot for your comments! We’ve already revised it, as shown in line 254.
« Line 262: does ‘significantly’ here mean ‘statistically significantly’? It would be good to include a test for statistical significance here. If you do not, please use another word to avoid confusion, such as ‘substantially’.
[Our Response]: Thanks a lot for your comments! We’ve already revised it, as shown in line 274.
« Figure 1: what is the feature importance score? I understand this is specific to XGBoost, but it should have a brief description since it is not commonly known.
Throughout section 2, please refer back to the RQs in section 2.1 to clarify which set of results corresponds to each research question.
[Our Response]: Thanks a lot for your comments! The explanation of feature importance score has been added in line 287-290: “When XGBoost is constructing a tree, it needs to split to make sequential classifications, and the feature importance score is judging the information gain of different features. The more information gains a feature can provide, the more likely it would be selected by XGBoost as the basis of classification, and the higher of the feature importance score.”
We agree it is advisable to refer back to the RQs in section 2.1 to increase the readability of the paper, we therefore refer back RQ1-RQ4 in the beginning of section 2.3.1, 2.3.3, 2.3.4 and 2.3.5 respectively.
« Line 282: this paragraph is very unclear. What do ‘above prediction process’, ‘under the same microblog’ and ‘current information’ refer to? I think you want to say something like “Consider the scenario where we have trained a model to predict spreaders. Now we observe a new rumor-refuting microblog and wish to find potential spreaders to recommend this microblog to. Is it possible to distinguish whether a user who was not in the training set will be a spreader for this microblog?”
[Our Response]: Thanks a lot for your comments! It is the same as what you state, and we made some adjustments about it, as shown in line 297-301: “This section answers RQ2 to check the generalizability of the model. Considering the scenario where we have trained a model to predict spreaders, now we observe a new rumor-refuting microblog and wish to find potential spreaders to recommend this microblog to. Is it possible to use the trained model for distinguishing even if the training set’s microblog topics are not exactly the same as the new rumor-refuting microblogs?”
« Line 296: what do you mean by robustness here, and how are you testing it? I see only F1 scores in the table, which do not vary much between the three subclass/major class setups.
[Our Response]: Thanks a lot for your comments! Since F1 scores were not sufficient to support the claim of robustness here, we removed the claim about robustness.
« Line 300-304: some of the text from the previous paragraph is repeated.
[Our Response]: Thanks a lot for your comments! We’ve already removed the repeated parts.
« Line 310-315: “the features were only learned from two sample microblogs before the trained model was used to predict”-- why is the training set different in this case to table 5? It should be trained on the same datasets to be comparable.
[Our Response]: Thanks a lot for your comments! In table 5, the training set and testing set may be extracted from different microblogs, the topic of their origin microblogs are of the same class (earthquake / man-made disasters like fire and explosion). However, in table 6, we did a more radical research by using training set and testing set from totally different microblog topics. We wonder if the XGBoost model is still efficient under this circumstance, thus we applied different datasets for table 5 and table 6.
« Lines 329 to 341: I cannot understand these paragraphs. Why would the PAR, PAC and PAL values be so different for spreaders and stiflers? What is this rescaling step that is needed and why? What does ‘heat’ mean in this context? Why does there need to be a new dataset rather than using the same dataset from 2.3.3 (RQ2)?
[Our Response]: Many thanks to your comments! We explained the motivation of considering group behavior variables PAR, PAC, PAL in line334-340. Thanks to your reminding, we realized the word ‘heat’ was confusing, so we revised it as ‘popularity’ as shown in line 355. The reason why we use a new dataset was that we tried to let the spreaders and stiflers of each microblog are within the same scale and proportion for the number of retweets/comments, as listed in line 348-354.
« Line 348-352: you say that group behavior did not contribute to identifying possible spreaders, but I think this is better phrased as “did not improve prediction of spreaders when added to the original set of features”. There is a correlation, and if you did not have the same set of original features, it is possible that group behavior would improve prediction.
[Our Response]: Thanks a lot for your comments! We’ve already revised it as shown in line 370-371.
« Line 361: After a major disaster, users may still be commenting on the disaster for more than a month. Therefore, it may not be realistic to use data from after the event as a substitute for data from before the event. This might explain the stronger performance on Microblog 1. Perhaps you can comment on this?
[Our Response]: Thanks a lot for your comments! Among the 4 rumors, only the disaster mentioned in Microblog2 actually happened (while largely exaggerated), and none of them (even the fabricated loss) is severe enough to make people continually talk about them after one month. Because different users have different tweet frequencies, it is technically difficult to get the microblogs of those users exactly before the rumor-refuting microblog for one month and several months. However, most users tweet much more than 10 microblogs within one month, and according to our data, few users’ microblog contents overlap their former contents collected before. Therefore, we think it is a good choice for us to use the contents after the rumor-refuting microblogs to simulate the contents before it.
« Lines 432 - 437: this seems to be just speculation and I don’t think it should be included in a scientific paper unless you provide some supporting evidence. It is quite off-topic as well: the paper is about detecting people who spread rumor-refuting blogs, yet this paragraph talks about the role of government and education. Many people would disagree that the government should interfere in social media, so this is a controversial topic that is probably too complex to mention briefly in passing.
[Our Response]: Thanks a lot for your comments! Indeed, it’s not objective and necessary to involve government measures in our thesis. Therefore, we deleted the content about government and education, instead we aimed to explain why detecting voluntary retweeters was very helpful in rumor-beating and how can this method be used to in the future in this paragraph, as shown in line 448-453.
« Typos and grammar:
« Line 13: natural language processing
« Lines 72/73: commas instead of semicolons; should it say ‘anti-rumor schemes’?
« Line 126: no semicolon needed
[Our Response]: Thanks a lot for your comments on the typos and grammar. Yes, it should be ‘anti-rumor schemes’. We revised the language and expressions carefully.
« Line 158: should ‘topping’ be ‘topic’ or ‘disaster topic’?
[Our Response]: Many thanks to your comments! The topping microblog, or the sticky microblog, is a function specific to the premium members of Sina Weibo. For a general user, the top microblog on his/her homepage is posted most recently, while for a premium member, he or she can choose one of his/her microblogs as the topping microblog, so you may see a microblog which was actually posted a long time ago on the top on his/her homepage.
« Line 212: full stop/period should be used instead of comma before ‘it is’.
« Line 235: should be “which uses the number of correctly predicted samples divided by the total sample size as a ratio,”
[Our Response]: Thanks for your comments! We revised these mistakes carefully. We have consulted a professional editor to help us to polish the whole paper.
« Table 9 -- not good to split across pages
« Line 384: ‘prediction’ not ‘prevision’?
« Line 385: ‘ranked highly’ not ‘ranked top’?
« Line 438: ‘Micro-post’ should be ‘micro-post’
[Our Response]: Thanks a lot for your comments! We reorganized our tables to avoid this problem, and revised the typos and grammars.
Although we have made a very thorough revision to the paper, there may be still some problems. We sincerely appreciate your further recommendations to enhance the quality of this paper.
Thank you very much and best regards.

Reviewer 2 Report
This research provides a general methodology for distinguishing disaster-related anti-rumor spreaders from a non-ignorant population base, with strong connections in their social circle. Several important influencing factors are also examined and illustrated. Real world dataset from microblog is used to evaluate the performances of different machine learning methods in predicting social media disaster rumor refuters. Generally speaking, the topic is interesting and the authors provided experiments on real world dataset. However, there are some critical issues as follows:
It is unclear that what factors are used in different machine learning methods and how the parameters are trained. More details should be provided in order for readers to understand the paper.
As authors mentioned, the aim of this research is to construct a model from a disaster related rumor refuting microblog so that the model can learn the spreader features and detect potential new spreaders. However, it is hard to find an overview and innovation of the model. It would be better if the authors can provide an overview and explain the innovation of the model. Maybe some figures are helpful.
Figure 1 and figure 2 are hard to read.
Author Response
Response to Reviewer 2
We have read the comments from you very carefully. We are grateful to you for your suggestions in improving this paper. Certainly, it has helped us to clarify several issues and hence, improved the paper.
We then give point-to-point response to the comments in the following, where the comments are marked with «.
« This research provides a general methodology for distinguishing disaster-related anti-rumor spreaders from a non-ignorant population base, with strong connections in their social circle. Several important influencing factors are also examined and illustrated. Real world dataset from microblog is used to evaluate the performances of different machine learning methods in predicting social media disaster rumor refuters. Generally speaking, the topic is interesting and the authors provided experiments on real world dataset. However, there are some critical issues as follows:
« It is unclear that what factors are used in different machine learning methods and how the parameters are trained. More details should be provided in order for readers to understand the paper.
[Our Response]: Thanks a lot for your comments, and we apologize that we didn’t make clear statements. We compared the efficiency of different machine learning methods by using the same factors. We applied the Python sklearn and XGboost packages to perform the model training, so the parameters training process are almost automatic except some hyperparameters need to be assigned and adjusted before training process to fit the data condition. We clarify the process as shown in line 226-227.
« As authors mentioned, the aim of this research is to construct a model from a disaster related rumor refuting microblog so that the model can learn the spreader features and detect potential new spreaders. However, it is hard to find an overview and innovation of the model. It would be better if the authors can provide an overview and explain the innovation of the model. Maybe some figures are helpful.
[Our Response]: Thanks a lot for your comments! We added a flowchart from data collection to data analyses, the process is now clearer in the flowchart. The innovation of this paper includes the introduction of text analysis as prediction factors in our machine learning classification model. To our best knowledge, prediction social media disaster rumor refuters has not been done before from individual perspectives. This is illustrated in line 190-191. We also highlighted the contributions in introduction (line 95-105).
« Figure 1 and figure 2 are hard to read.
[Our Response]: Thanks a lot for your comments! We improved the resolution of Figure 1 and Figure 2 (they are now Figure 2 and Figure 3) to make them clearer.
Although we have made a very thorough revision to the paper, there may be still some problems. We sincerely appreciate your further recommendations to enhance the quality of this paper.
Thank you very much and best regards.

Reviewer 3 Report
This study investigated the performance of machine learning techniques in predicting social media disaster rumor refuters. First, nature language processing was employed for sentiment and short text analysis. Then, four machine learning methods including logistic regression, support vector machines, random forest and extreme gradient boosting were considered as the prediction models. Finally, the capacities of four models were evaluated and compared based on microblog. Overall, the topic of the manuscript is interesting and the structure is well organized. My comments are provided as follows.
1. Introduction, please clearly state the motivation for selecting these four models for the tasks of interest. Why not neural network, gene expression programming, or deep learning models?
2. More references within latest 5 years should be added in the introduction. I recommend the following references for better illustration of SVM method.
https://doi.org/10.1016/j.conbuildmat.2018.06.219
https://doi.org/10.1016/j.neucom.2016.02.074
https://doi.org/10.1088/0964-1726/24/3/035025
3. The flowchart/schematic about the ML methods to predict the social media disaster rumor refuters should be included in Section 2.
4. The authors should better illustrate how the data were collected and selected and how to filter out the outliers.
5. Most ML methods have the drawback of overfitting problem. How do the authors avoid this problem in this study?
6. More future research should be included in the Conclusions part.
7. There are several typos that can affect the quality of the manuscript. Please revise them.
Author Response
Response to Reviewer 3
We have read the comments from you very carefully. We are grateful to you for your suggestions in improving this paper. Certainly, it has helped us to clarify several issues and hence, improved the paper.
We then give point-to-point response to the comments in the following, where the comments are marked with «.
« This study investigated the performance of machine learning techniques in predicting social media disaster rumor refuters. First, nature language processing was employed for sentiment and short text analysis. Then, four machine learning methods including logistic regression, support vector machines, random forest and extreme gradient boosting were considered as the prediction models. Finally, the capacities of four models were evaluated and compared based on microblog. Overall, the topic of the manuscript is interesting and the structure is well organized. My comments are provided as follows.
« 1. Introduction, please clearly state the motivation for selecting these four models for the tasks of interest. Why not neural network, gene expression programming, or deep learning models?
[Our Response]: Thanks a lot for your comments! We applied logistic regression and SVM because they are generally used as benchmarks in machine learning researches. Our focuses are random forest and XGBoost, two tree-structured algorithms proven to have great performance of accuracy in classification. Neural network can get very good classification results, but the hidden processes are hard to explain while we also need to implement feature importance analysis, it is not quite fit. As for deep learning, and since two of the microblog datasets only get hundreds of samples, it is not enough for deep learning models to obtain good classification results. We added more description about selection of models in line 111-113.
« 2. More references within latest 5 years should be added in the introduction. I recommend the following references for better illustration of SVM method.
https://doi.org/10.1016/j.conbuildmat.2018.06.219
https://doi.org/10.1016/j.neucom.2016.02.074
https://doi.org/10.1088/0964-1726/24/3/035025
[Our Response]: Thanks a lot for your comments! Theses literature really inspires us, we have added the 3 references in Section 2.3.1 of our revised paper, as shown in Line 215-218.
« 3. The flowchart/schematic about the ML methods to predict the social media disaster rumor refuters should be included in Section 2.
[Our Response]: Thanks a lot for your comments! We have added a flowchart to show the process more clearly, as shown in Figure 1 of revised paper.
« 4. The authors should better illustrate how the data were collected and selected and how to filter out the outliers.
[Our Response]: Thanks a lot for your comments! For this dataset, we don’t have the problems of outliers, as the data we get from the similarity and sentiment analyses are between [0,1] naturally. All the individual information is worthy of including since there are all kinds of netizens in reality. Besides, the LR and SVM model need to standardize the data before training, thus there is no extreme value for them. For RF and XGBoost, there is no need for data standardization since the tree structure can handle them well. To make the data collection clearer, we polished section 2.2.
« 5. Most ML methods have the drawback of overfitting problem. How do the authors avoid this problem in this study?
[Our Response]: Thanks a lot for your comments! We tried to avoid overfitting by assigning suitable hyperparameters. For instance, for SVM, we set a relatively small penalty coefficient, thus the precision on the training data set may decrease while the generalization on the testing data will increase. For RF and XGBoost, we avoid constructing the tree to be too deep, and we adjust the eta coefficient, which can contract the weight of features to make the improving process more conservative. We checked the fitting results, it did not show any evidence of overfitting, namely well-fit in trainning data and poorly-fit in the test data.
« 6. More future research should be included in the Conclusions part.
[Our Response]: Thanks a lot for your comments! Considering the drawbacks of only choosing four disaster events, it’s necessary to cover larger number of disaster events and more types of various sample in the future work. In the revised conclusions We acknowledged the limitations and promised to increase the sample size and use high subdivision sample in future work, as shown in the line 478-480: “Choosing more disaster events and larger sample size are also needed to illustrate the generality of our results. More broadly, the study could extend into various social media platforms to enhance applicability of our model and optimize the method.”
« 7. There are several typos that can affect the quality of the manuscript. Please revise them.
[Our Response]: Thanks a lot for pointing out the language issue. We have consulted professional editor to help us polish the whole paper regarding language and expressions.
Although we have made a very thorough revision to the paper, there may be still some problems. We sincerely appreciate your further recommendations to enhance the quality of this paper.
Thank you very much and best regards.

Round 2
Reviewer 3 Report
The authors carefully revised the manuscript and well addressed the reviewer comments. Accordingly, I suggest that current version of manuscript can be published in IJERPH.